# Woody Plant Encroachment in a Seasonal Tropical Savanna: Lessons about Classifiers and Accuracy from UAV Images

**Lucas Silva Costa** [1,*] **, Edson Eyji Sano** [2] **, Manuel Eduardo Ferreira** [3] **, Cássia Beatriz Rodrigues Munhoz** [1] **, João Vítor Silva Costa** [3] **, Leomar Rufino Alves Júnior** [3] **, Thiago Roure Bandeira de Mello** [1] **and Mercedes Maria da Cunha Bustamante** [1]

1  Programa de Pós-Graduação em Ecologia, Instituto de Ciências Biológicas, Universidade de Brasília, Brasília 70910-900, Brazil
2  Empresa Brasileira de Pesquisa Agropecuária (Embrapa Cerrados), BR-020, Planaltina 73301-970, Brazil
3  Instituto de Estudos Socioambientais, Universidade Federal do Goiás (UFG), Goiânia 74690-900, Brazil
*  Correspondence: lucas.costa@aluno.unb.br

**Abstract:** Woody plant encroachment in grassy ecosystems is a widely reported phenomenon associated with negative impacts on ecosystem functions. Most studies of this phenomenon have been carried out in arid and semi-arid grasslands. Therefore, studies in tropical regions, particularly savannas, which are composed of grassland and woodland mosaics, are needed. Our objective was to evaluate the accuracy of woody encroachment classification in the Brazilian Cerrado, a tropical savanna. We acquired dry and wet season unmanned aerial vehicle (UAV) images using RGB and multispectral cameras that were processed by the support vector machine (SVM), decision tree (DT), and random forest (RF) classifiers. We also compared two validation methods: the orthomosaic and in situ methods. We targeted two native woody species: *Baccharis retusa* and *Trembleya parviflora*. Identification of these two species was statistically ($p < 0.05$) most accurate in the wet season RGB images classified by the RF algorithm, with an overall accuracy (OA) of 92.7%. Relating to validation assessments, the in situ method was more susceptible to underfitting scenarios, especially using an RF classifier. The OA was higher in grassland than in woodland formations. Our results show that woody encroachment classification in a tropical savanna is possible using UAV images and field surveys and is suggested to be conducted during the wet season. It is challenging to classify UAV images in highly diverse ecosystems such as the Cerrado; therefore, whenever possible, researchers should use multiple accuracy assessment methods. In the case of using in situ accuracy assessment, we suggest a minimum of 40 training samples per class and to use multiple classifiers (e.g., RF and DT). Our findings contribute to the generation of tools that optimize time and cost for the monitoring and management of woody encroachment in tropical savannas.

**Keywords:** Cerrado; object-based image analysis; mesic biome; plant invasion; drone; multispectral; machine learning; grasslands; woodlands; in situ ground truth

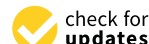



## 1. Introduction

Plant invasions are an increasing challenge for the management of native biodiversity and ecosystem functioning worldwide. Invasive plants establish themselves in habitats and proliferate, spread, and persist to the detriment of other species and overall environmental conditions [1]. Biological invasions can be associated with both exotic species and local species that increase significantly in abundance [2]. Invasive species can displace or promote the extinction of resident species and alter biogeochemical cycles, energy flux, and disturbance regimes [1,3,4].

The invasion of woody plants is also referred to as woody plant encroachment. Following the concept proposed by Irini and Chui [5]; here we also consider woody encroachment as the invasion of native woody plants. Woody plant encroachment has been widely

reported, mainly in grasslands and open woodlands [6–8]. Van Aucken [9] associated this phenomenon with increased population density, biomass, or the land cover of native woody plants.

Savannas are composed of a continuous grass–subshrub layer with scattered woody plants. Rainfall seasonality, fire occurrence, and low soil nutrient availability are factors that allow the co-occurrence of tree–grass layers [10,11], mainly in tropical savannas. Accordingly, factors behind woody encroachment include fire suppression, nutrient eutrophication, changes in water dynamics, increased $CO_2$ emission, land use conversion, and livestock overgrazing [12–14].

Woody encroachment is associated with negative impacts on ecosystem processes and functions [9,15]. For example, the increase in and homogenization of woody layers alter spatial patterns of geochemical properties, such as the formation of "fertile islands" below the canopies and scarcity zones in the intra-canopy spaces [16–18]. Additional reported consequences include decreased biodiversity in the grass/herbaceous layer [9,19–23] and impacts on the water cycle and climate [12,24,25]. Initial environmental changes trigger woody encroachment, and a positive feedback loop can continue altering the environment past the tipping point, thereby pushing ecosystems to a new equilibrium [26]. This interaction represents a challenge for the conservation of areas under land use change and protected areas [27,28].

In a scenario of rapid environmental change, efforts to conserve and manage natural areas must be optimized. Nackley et al. [29] suggested that ecologists and natural resource managers integrate empirical evidence that quantifies the impacts of woody encroachment with their natural resource management strategies, especially in understudied regions. Currently, our understanding of woody encroachment is based mainly on studies in arid and semi-arid regions of the African continent [14,17,30–32], Oceania [33,34], Mediterranean countries [35,36], and North America [9,15,18,28,37]. Tropical regions, especially in South America, remain understudied.

Savannas occupy about 40% of the land surface in the tropics. The Brazilian Cerrado constitutes the largest savanna in South America and is considered one of the world's biodiversity hotspots and a priority for conservation because of its outstanding biodiversity and extensive loss of natural habitats for agricultural use [38,39]. However, the Cerrado is less protected than the Amazon by law, with conservation units and indigenous lands accounting for only 11% of its original area [40]. The vegetational mosaics, from forests to grassland formations [41], create a structural heterogeneity that is one of the factors sustaining its megadiversity. However, woody encroachment tends to homogenize these formations, especially the grasslands. The management of conservation units can be improved with better detection of woody plant encroachment, which can be accomplished with low-cost technologies.

The challenges of early detection and prevention of new biological invasions [42] could be overcome by implementing monitoring options that optimize time and cost and provide decision makers with precise and accurate information in the most efficient way possible. Thanks to its synoptic and temporal coverage, remote sensing data have become increasingly important for ecological monitoring [43], understanding factors that promote plant invasions and their processes [44], and assessing impacts on functional attributes and ecosystem services [45,46]. Therefore, remote sensing technology has been recommended to control woody encroachment by some studies, e.g., [36,47,48], particularly within protected areas. Because of its favorable cost–benefit ratio, the use of unmanned aerial vehicle (UAV)-based data has emerged as a useful approach to detect and monitor the encroachment process.

Monitoring the environment from UAV-based platforms is now relatively common. The costs of current technologies have allowed UAVs to be an essential data source for multiple applications [49]. Despite the improved temporal and spatial resolution, satellite-based remote sensing is still limited to monitoring and identifying individual plant species [49,50]. In contrast, UAVs generate data with sub-metric resolution, which enables the identifi-

cation, classification, and monitoring of woody encroachments [36,47,48,51,52]. UAVs can be equipped with RGB, multispectral, hyperspectral, and active sensors. Although Olariu et al. [52] presented satisfactory results using only RGB cameras, multispectral and hyperspectral sensors can also be used to map vegetation [51,53,54], especially in wet grasslands [55]. However, there is a lack of studies combining RGB and multispectral sensors for this purpose. RGB cameras generate 3D point clouds as an additional product, allowing the extraction of vertical structure information, which can improve classification procedures if combined with multispectral bands. Dealing with complex vegetation (highly biodiverse), we tested different degrees of complexity of input layers. Although additional spectral and structural information can improve the final classification, the combination of sensors demands high data processing capability.

Machine learning algorithms are widely used tools in remote sensing classification. They can model complex class signatures and accept different types of input data, including data without normal distribution. There is sufficient evidence of the superior performance of machine learning algorithms over traditional techniques, such as maximum likelihood estimation [56–59]. Some authors have suggested using more than one classifier to minimize possible classification biases [60]. For example, Maxwell et al. [59] recommended the following machine learning algorithms: support vector machine (SVM), decision tree (DT), and random forest (RF).

Although a well-established algorithm, the SVM has presented controversial results in relation to assembly classifiers such as RF, sometimes with superior [61–63] and sometimes with inferior accuracy [64,65]. The same need for future comparative studies occurs with DT; despite RF being composed of an ensemble of DTs, the results provided by the DT are much easier to interpret and can be built from direct inspection of the variable, while RF is much more complex in terms of parameter predefining, sample training, and result voting, as stated by [66]. Moreover, studies conducted, for example, by [67] and [68], showed that DT overperformed artificial neural network (ANN) and SVM algorithms. Based on these studies, and in line with Maxwell et al. [59], SVM, DT, and RF are worth including in our study.

These powerful classifiers produce high overall accuracies (OAs), especially for complex data with many predictor variables. Although deep learning algorithms can outperform machine learning [69,70], they require in-depth knowledge of programming and high processing capability, resulting in higher costs. Here, we chose to use machine learning because of its robustness. Furthermore, machine learning is user-friendly and widely available on public domain platforms, which is important to help natural resource managers deal with woody encroachment in environmental protection units.

Our overall objective was to evaluate the classification accuracy of woody encroachment in a tropical savanna by comparing data obtained by RGB and multispectral cameras onboard a UAV platform. We obtained data during the wet and dry seasons in two grassland formations and two savanna woodland formations. Using three widely used classifiers (SVM, DT, and RF), we hypothesized that we would be able to obtain accurate classifications, especially in grasslands, because our two focal species (*Trembleya parviflora* and *Baccharis retusa*) are shrubs, which are more easily detected in grasslands. We also hypothesized that the best input layer (combination of spectral bands, spectral indexes, and metrics) would be the one that combines the RGB and multispectral cameras, which can provide broader spectral coverage, better differentiating the spectral signatures of each class.

## 2. Materials and Methods

### 2.1. Study Area

The study was carried out in the Botanic Garden of Brasília (JBB), located 33 km from the center of Brasília, Federal District, Brazil (Figure 1). The JBB is part of the Gama-Cabeça-de-Veado Environmentally Protected Area, which is one of the three largest conservation units in the Federal District. The climate is classified as Aw in the Köppen

climate classification system; that is, tropical with dry winter and rainy summer. The average annual rainfall is approximately 1500 mm. The study area (15°52′47.49″ south latitude; 47°51′12.87″ west longitude) covers an area of approximately 20 ha, encompassing five Cerrado phytophysiognomies: campo úmido (wet grasslands), campo sujo (grasslands with scattered woody plants), cerrado ralo (open savannas), cerrado típico (typical Brazilian savanna), and cerrado rupestre (rocky savanna) [41]. Our study was carried out in two grasslands (campo úmido and campo sujo) and two savanna woodlands (cerrado ralo and cerrado típico). The soils in the study areas were mainly sandy clay loam, sandy loam, and loamy sand. This reserve was chosen because *B. retusa* and *T. parviflora* have colonized the savanna woodlands and grasslands, and there is a need to manage these species.

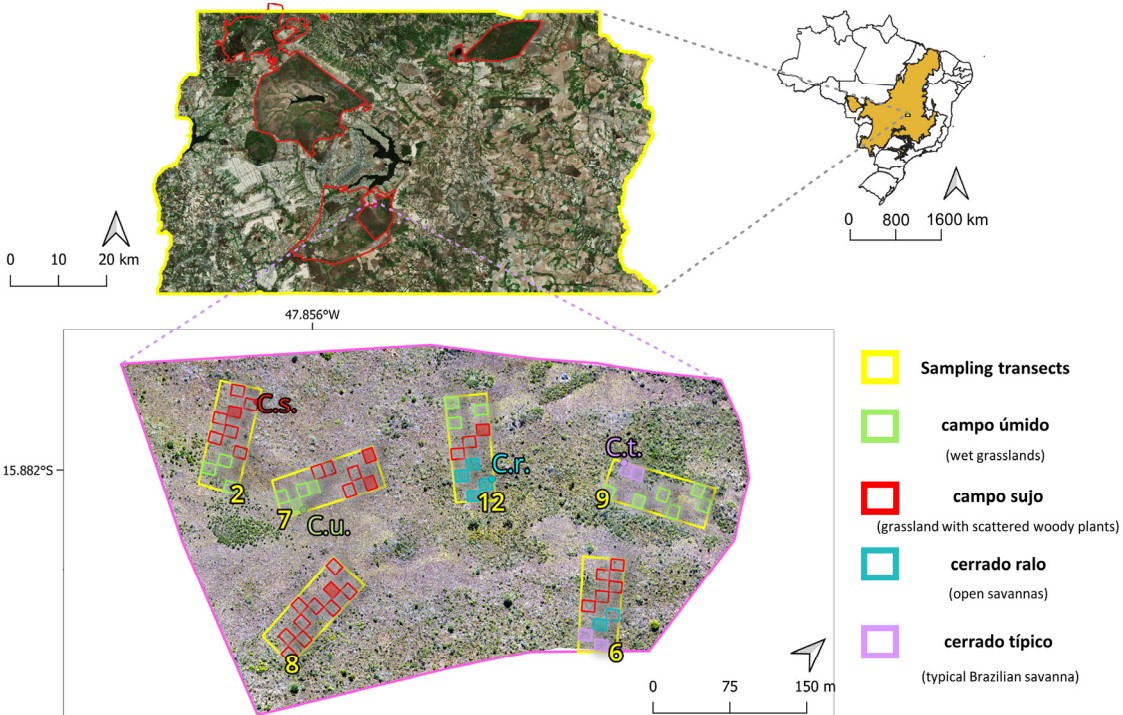

**Figure 1.** The study area location. In the upper right map, the Cerrado area is represented by orange shading, and the yellow square indicates the borders of the Federal District (DF), Brazil. In the upper left map, red lines indicate the edges of the three largest conservation units in the DF, and the pink polygon indicates the drone flight area. In the lower map, blurred colored quadrants indicate the quadrants randomized for the in situ validation of the classification maps.

## 2.2. Studied Species

The *Baccharis L.* genus is diverse, with 440 species [71], of which 120 occur in Brazil. The species *B. halimifolia* is studied on the North American, European, and Oceania continents, where it is considered invasive and a priority for control and management because of its environmental and economic impacts [72]. *B. pilularis* and *B. spicata* are monitored mainly in Europe because of their invasive potential [73,74], and *B. dracunculifolia* is a potential invader in 33 countries from five continents. The best way to counter the invasions is prevention, followed by monitoring and management [75]. *B. retusa*, as with the congeners mentioned above, is classified as a generalist and is adapted to pioneer stages of succession. This species is native to the Cerrado, occurring mainly in woodlands.

*Trembleya parviflora* is an erect shrub, endemic to Brazil and native to the Atlantic Forest and Cerrado biomes. *T. parviflora* occurs in campo sujo, campo limpo, cerrado rupestre, veredas, and the edges of the riparian and gallery forests. Its fruits contain many tiny seeds that are dispersed by the wind in August and September. In the Federal District, rapid

landscape transformation is occurring because of dense *T. parviflora* colonization, resulting in the loss of the native herbaceous community in wetlands [23].

### 2.3. Field Data Gathering and Digital Image Processing

The UAV imageries were acquired between 12:00 and 13:00 local time on 17 July 2021 (dry season) and between 13:00 and 14:00 on 4 April 2022 (wet season) with a DJI Phantom 4 standard drone carrying out a Sony 1/2.3″ CMOS RGB camera with a 94° field of view, and 20 mm lens. A total of 700 images were collected at a flying height of 50 m, generating a 2.80 cm average ground sampling distance (GSD). We used two different UAVs to also acquire multispectral imagery. During the dry season, we used a Parrot Bluegrass drone equipped with a Sequoia 4.0 sensor (green, red, red edge, and near-infrared (NIR) bands), generating 2532 images with a 7.07 cm GSD. During the wet season, we used a DJI Matrice 200 v2 drone with an Altum sensor (green, red, blue, red edge, NIR, and longwave infrared (LWIR/Thermal) bands), resulting in 9474 images with a 2.99 cm GSD. Only the four bands also available for images' dry period were used for comparisons between the two seasons. All flights had support from geodetic global navigation satellite system (GNSS) ground points to increase planimetric and altimetric accuracies. As a result, we generated 14 geometrically corrected orthomosaics using the Pix4Dmapper software (v. 4.7), seven for each season, in addition to two 3D point clouds, the digital surface models (DSMs), and digital terrain models (DTMs).

### 2.4. Generation of Layers, Masks, and Metrics

All RGB and multispectral bands were resampled to a spatial resolution of 10 cm to standardize all GSDs and minimize data processing time. Because the work focused on two shrub species, we applied a mask limiting only vegetation above 50 cm from the ground. This mask was obtained from the canopy height model (CHM) (Equation (1)).

$$\text{Canopy height model (CHM)} = \text{DSM} - \text{DTM} \tag{1}$$

We generated two spectral indices based on RGB data to improve classification, as suggested by Olariu et al. [52]: green–red difference and green leaf index referred to as IDXRGB (Equations (2) and (3), respectively). Two other indexes were generated based on a multispectral camera, applying NIR and red edge bands: normalized difference vegetation index (NDVI) and normalized difference red edge (NDRE) index (Equations (4) and (5), respectively), referred to as IDXMult hereafter (Table 1).

$$\text{Green–red difference} = \frac{(\text{green band} - \text{red band})}{(\text{green band} + \text{red band})} \tag{2}$$

$$\text{Green leaf index} = \frac{(2 \times \text{green band} - \text{red band} - \text{blue band})}{(2 \times \text{green band} + \text{red band} + \text{blue band})} \tag{3}$$

$$\text{NDVI} = \frac{(\text{NIR band} - \text{red band})}{(\text{NIR band} + \text{red band})} \tag{4}$$

$$\text{NDRE} = \frac{(\text{NIR band} - \text{red edge band})}{(\text{NIR band} + \text{red edge band})} \tag{5}$$

We derived textural information from the RGB orthomosaics, calculating gray-level co-occurrence matrix (GLCM) metrics. The texture metrics were calculated using the average values of the RGB bands and included the following eight textural features: energy, entropy, correlation, inverse difference moment, inertia, cluster shade, cluster prominence, and Haralick correlation. We extracted the textural metrics with the Orfeo ToolBox (OTB) plugin available in the QGIS software (version 2.22) using the "HaralickTextureExtraction" function. Orfeo ToolBox (OTB). Developed by the French Center National d'Etudes Spatiales, OTB can be operated either autonomously or through a second open-source

software (QGIS). OTB uses the C++ library, based on the Insight toolkit (ITK). Bindings are developed for Python.

**Table 1.** Summary of UAV-based predictor types, their abbreviations, and respective number of bands. NDVI refers to the normalized difference vegetation index, and NDRE refers to the normalized difference red edge index.

| Predictors | Abbreviation | No. Bands |
|---|---|---|
| Canopy height model | CHM | 1 |
| Red, green, and blue bands | RGB | 3 |
| Texture | Text | 8 |
| Structure | Stru | 10 |
| Green, red, red edge, and NIR bands | Mult | 4 |
| Green leaf index and green–red difference | IDXRGB | 2 |
| NDVI and NDRE | IDXMult | 2 |
| Six principal components of all bands | PCA | 6 |

Additionally, we calculated structural metrics by applying the CANUPO multiscale component analysis [51,76] using the dense point cloud. The calculation of the structural metrics is based on principal component analysis (PCA) of the 3D neighborhood of the respective spatial scale, where each metric corresponds to the difference between the first and second normalized eigenvectors [76]. Ten spatial scales ranging from 10 cm to 3 m were chosen based on the structural parameters of the canopies of the two focal species. This allowed detection of each species' canopy characteristics, from branch arrangement to leaf phyllotaxis. The combination of the CANUPO analysis in the CloudCompare software and the cloth simulation filter (CSF) plugin was used to discriminate soil from vegetation [70] and thus provided two distinct classes for use as input in the training and classification of CANUPO. As suggested by Kattenborn et al. [51], only the uppermost canopy points were used to calculate the output raster to avoid interference with non-canopy characteristics.

We derived the last layer from all 30 available bands (Table 1). We performed the data dimensionality reduction using the "DimensionalityReduction" function in the OTB plugin, also available in QGIS. Applying PCA as the dimensionality reduction method, we also chose six components as output to have the same number of bands as the smallest possible combination of bands (e.g., RGB + IDXRGB + CHM = 6 bands). PCA is the most common approach to dimensionality reduction. Since we targeted a larger number of end users, we chose a technique that is widely used and is available in largely open-source software packages.

To stack the predictors and make the desired combinations, we aligned raster files using the QGIS plugin Freehand Raster Georeferencer plugin. To align the predictors, we used the two-point alignment technique, with two of the points (ground targets) used to calibrate geodesic GNSS. In the end, we produced seven combinations from eight predictors (Table 1). Three combinations used mainly RGB bands: RGB + IDXRGB + CHM; RGB + IDXRGB + CHM + Text; and RGB + IDXRGB + CHM + Stru. In the same way, we generated three combinations using mainly multispectral bands: Mult + IDXMult + CHM + Text; Mult + IDXMult + CHM + Stru; and Mult + Text + Stru. Finally, we considered the PCA layer separately as it represents the derivation of all bands.

### 2.5. Image Segmentation and Zonal Statistics

To minimize the "salt and pepper" effect common in higher-resolution images from UAVs, we performed object-based image analysis, as suggested by Olariu et al. [52], which required segmentation and subsequent calculation of zonal statistics. We used the segmentation function available in the OTB plugin associated with QGIS software with the mean shift algorithm for all layers. Although Olariu et al. [52] showed that classification was improved with large-object segmentation (e.g., minimum region size = 100), we generally

chose a minimum region size of 30 to include young individual shrubs in the classification. The exception was the Mult + IDXMult + CHM + Text layer, which used a minimum region size of 200 because the objective here was to obtain a better representation of the shrub's canopy as a unit. For the other parameters, we kept the default values for all layers (spatial segmentation ratio = 5 and range radius = 15). We applied the calculation of zonal statistics over the segmentation vectors. For this, we used the model designer function in QGIS for all layers, being necessary only for the adaptation of the model for the number of bands of each layer. We chose the following nine parameters to calculate zonal statistics: mean, median, standard deviation, minimum, maximum, range, minority, variability, and variance. Aiming for a more normal data distribution of the zonal statistics, three of the 12 metrics were removed, they were: count, sum, and variety. Before performing the analysis, we carried out pilot tests to assess the influence of these metrics on the accuracy of the models, and it was finally decided to remove them for all models. The reason was that they had a very discrepant scale compared with the other metrics.

### 2.6. Object-Based Supervised Classification

The following three classifiers that are widely used in vegetation mapping [77–89] were used in this study, with some differences in approach: SVM, DT, and RF. SVM [80] is a non-parametric method, so it does not rely on data normality and aims to find the optimal limit that maximizes the separation between classes. The SVM classifier works by identifying a unique boundary between two classes. In multiple class cases, SVM repeatedly applies the classifier to each possible combination of classes. One of its main limitations is processing time, which rises exponentially as the number of classes increases [59,80].

DT [56] is a recursive division of the input layers, where the data can be divided depending on whether the value is above or below a threshold. The tree analogy describes repeated division patterns (e.g., branch vertices). DTs can use both categorical and continuous data. One of its advantages is that, once the model has been developed, classification is rapid, because no further complex calculations are required. However, there is a possibility of generating non-optimal solutions and overfittings [56,59].

RF [81] is an ensemble classifier because it uses multiple DTs to overcome the limitations of a single DT. The majority result of all DTs is used to define the final class, requiring the need to obtain the global optimum. An advantage of RF is that individual trees do not need to be pruned because of the multiple DTs. A disadvantage is that the ability to view individual trees is lost [59,81].

We used the OTB plugin in QGIS to generate all classification models through the TrainVectorClassifier function. We used the polynomial Kernel, a C support vector classification as SVM mode, and a degree parameter equals three in the SVM. The final RF classification used 1000 decision trees with a tree depth of 50. The same maximum tree depth was used for the DT classifier.

We selected five land cover classes: *T. parviflora* (T.P), *B. retusa* wood (B.Rw), *B. retusa* green (B.Rg), shadow (Sh), and others (Ot). We separated *B. retusa* into two classes because the species is semi-deciduous and there was a considerable amount of standing dead materials; therefore, the canopy included a significant portion of leafless wood (B.Rw). Although most of the shadows were excluded with the CHM mask, we noted that a considerable portion of the shadow was classified as a plant. Finally, for highly diverse phytophysiognomies (i.e., woodlands), which included individuals taller than 50 cm, we added the category of "others" to represent any other species.

Additionally, we used the McNemar test to verify the statistical significance among the different classifiers. Since the test requires paired samples and only allows two-by-two comparison, we chose only the best results among the RGB, multispectral, and PCA predictors, as well as for the season whose classifiers presented the best performances. We ran the McNemar without Yates' correction test due to sample size.

We analyzed the learning curves to compare the different algorithm performances, specifically for the orthomosaic data validation, and evaluated the number of sampling data

used for training in situ validation. In both purposes of learning curves, we used the zonal statistics of each input layer, with the difference being that for the in situ validation method, we used a group of only 40 selected zonal statistics (10 per class). The learning curves were made in the Jupyter Notebook interface using the Numpy and Sklearn (Python) libraries.

### 2.7. Accuracy Assessment and In Situ Validation

To use the same canopies for the training and validation of all classifiers and input layers, thus reducing external variability, 100 points were stipulated for each class. We used the function "Select From Location" available in QGIS so that most of the points would select essentially the same canopies to allow 70% of the data for training and 30% for validation. Minor adjustments were necessary because the segmentation differed between input layers. Each segment has the structure of polygons grouped by the similarity of pixels during the segmentation process using the mean shift algorithm. In the end, we used the training and validation selected segments as input on the "TrainVectorClassifier" function, resulting in the model and confusion matrix files. We used the model file to generate classification maps with the "VectorClassifier" OTB function.

Additionally, we performed in situ validation of the classification maps to assess whether accuracy differed according to phytophysiognomy. A set of 12 transects of 100 m × 30 m was arranged perpendicular to the drainage network, and six transects were randomly selected for a focal species. Additionally, for every 10 m in a randomized quadrant of 10 m × 10 m, we identified the vegetation phytophysiognomy (Figure 1). Because only four of the quadrants were classified as cerrado típico, four quadrants were randomly chosen for each of the other three phytophysiognomies. To optimize fieldwork, we chose the three input layers for which the highest average OA and Kappa coefficient were obtained in both seasons. We preferred having one input layer from the RGB sensor and one from the multispectral sensor: RGB + RGB + CHM; Mult + Text + Stru and PCA.

For in situ validation, we randomized approximately 10 canopies of each class (Vector → Research → Random Selection) within each phytophysiognomy, for approximately 440 canopies in all. These reference canopies were highlighted and extracted from the classification maps. We generated 54 georeferenced PDF files used in the field to validate each of the canopies via mobile device with the help of the Avenza Maps® (v.4.1) application. The in situ validation protocol included locating the center of the permanent quadrants, setting up the 10 m x 10 m quadrant, and filling the confusion matrix with the help of the georeferenced PDFs. Additionally, we stipulated a 2 m radius buffer zone around each classified canopy to improve mobile GPS accuracy. In the end, it was possible to generate two accuracy assessments, one broad and another by phytophysiognomy. The workflow is summarized in Figure 2.

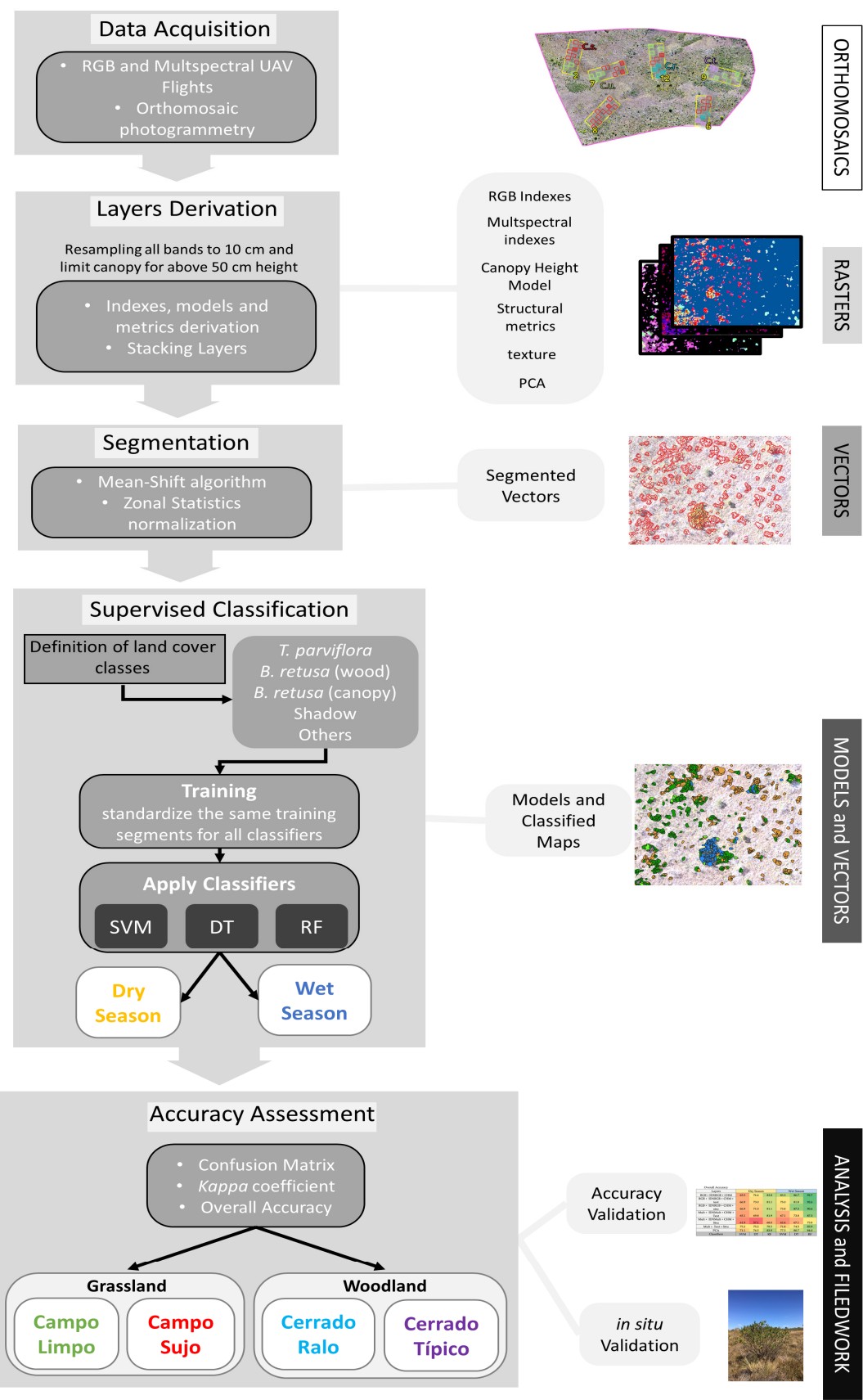

**Figure 2.** Workflow diagram illustrating the UAV-based image processing, classification, and accuracy assessment.

## 3. Results

### 3.1. Classifiers, Input Layers, and Climatic Seasons' Accuracy Assessment

The overall accuracy (OA) of the 42 combinations of input layers and classifiers, performed in the dry and wet seasons, ranged from 57.1% to 92.7%, and the Kappa measure of agreement ranged from 0.46 to 0.91 (Tables 2 and S1). The RF classifier obtained the most accurate result with the RGB + IDX + CHM layer for the wet season and was statistically ($p < 0.01$) superior to the performance of SVM. The RF classifier also performed statistically ($p < 0.01$) better than the other classifiers (DT and SVM) when using Mult + IDXMult + CHM + Text as a predictor (Tables 2 and S1). Using PCA as a predictor, no significant difference was found among the results according to the McNemar test. In general, the input layers generated from orthomosaics during the wet season were more accurate.

**Table 2.** Overall accuracies (AOs) and Kappa coefficients are displayed for two seasons (dry and wet), seven input layers, and three machine learning classifiers. For abbreviation identification of input layers, see Table 1. Lowercase letters indicate statistically significant difference ($p < 0.01$) according to the McNemar test among classifiers.

| Layers | Dry Season | | | Wet Season | | |
|---|---|---|---|---|---|---|
| RGB + IDXRGB + CHM | 65.5 (0.57) | 76.4 (0.70) | 83.8 (0.80) | 83.3 [a] (0.79) | 86.7 [ab] (0.83) | 92.7 [b] (0.91) |
| RGB + IDXRGB + CHM + text | 66.9 (0.59) | 73.0 (0.66) | 81.1 (0.76) | 75.0 (0.69) | 81.8 (0.77) | 92.6 (0.91) |
| RGB + IDXRGB + CHM + stru | 66.9 (0.60) | 71.0 (0.64) | 81.1 (0.76) | 73.8 (0.67) | 87.3 (0.84) | 90.6 (0.88) |
| Mult + IDXMult + CHM + Text | 65.1 (0.56) | 69.8 (0.62) | 81.9 (0.77) | 67.1 [a] (0.59) | 73.8 [a] (0.67) | 87.3 [b] (0.84) |
| Mult + IDXMult + CHM + Stru | 61.9 (0.52) | 57.1 (0.46) | 68.0 (0.60) | 62.4 (0.53) | 67.1 (0.60) | 75.8 (0.70) |
| Mult + Text + Stru | 75.2 (0.69) | 75.2 (0.69) | 78.5 (0.73) | 71.8 (0.65) | 74.5 (0.68) | 85.9 (0.82) |
| PCA | 71.1 (0.64) | 76.5 (0.70) | 83.9 (0.80) | 77.3 (0.72) | 80.7 (0.76) | 84.0 (0.80) |
| Classifiers | SVM | DT | RF | SVM | DT | RF |

Among the evaluated models, the average OA and agreement were the highest for the RF classifier, and among the input layers, the average OA was highest for RGB + IDX + CHM, followed by PCA and RGB + IDX + CHM + Stru. Among the classes of interest (i.e., *T. parviflora* and *B. retusa*), the lowest commission and omission errors were found for *T. parviflora*, followed by *B. retusa* (wood), using the RF classifier (Tables 3, 4 and S1). As expected, the additional classes, Sh and Ot, were associated with the highest and lowest user and producer accuracies, respectively, for all classifiers. A comparison of the two *B. retusa* classes showed that the classification accuracy of leafless wood (B.Rw) was better than that of the green canopy (B.Rg) (Tables 3, 4 and S1).

**Table 3.** User's (UA) and producer's accuracy (PA) for each class of land cover: *Trembleya parviflora* (T.P.); *Baccharis retusa*—wood (B.Rw); *Baccharis retusa*—green canopy (B.Rg), Shadow (Sh), and Others (Ot) in two seasons (wet and dry) using three machine learning classifiers. For abbreviation identification of input layers, see Table 1.

| Classifier | Support Vector Machine | | | | | | | | | | Decision Tree | | | | | | | | | | Random Forest | | | | | | | | | |
|---|---|---|---|---|---|---|---|---|---|---|---|---|---|---|---|---|---|---|---|---|---|---|---|---|---|---|---|---|---|---|
| Class | T.P | | B.Rw | | B.Rg | | Sh | | Ot | | T.P | | B.Rw | | B.Rg | | Sh | | Ot | | T.P | | B.Rw | | B.Rg | | Sh | | Ot | |
| Wet Season/Layer | UA | PA | UA | PA | UA | PA | UA | PA | UA | PA | UA | PA | UA | PA | UA | PA | UA | PA | UA | PA | UA | PA | UA | PA | UA | PA | UA | PA | UA | PA |
| RGB + IDXRGB + CHM | 88 | 78 | 81 | 93 | 69 | 73 | 98 | 95 | 79 | 76 | 96 | 82 | 87 | 79 | 81 | 87 | 97 | 97 | 71 | 87 | 100 | 80 | 97 | 88 | 81 | 98 | 99 | 97 | 86 | 99 |
| RGB + IDXRGB + CHM + text | 91 | 85 | 78 | 82 | 63 | 66 | 78 | 90 | 63 | 55 | 91 | 91 | 75 | 82 | 87 | 68 | 83 | 91 | 74 | 83 | 99 | 91 | 99 | 86 | 93 | 93 | 87 | 98 | 78 | 99 |
| RGB + IDXRGB + CHM + stru | 83 | 89 | 52 | 88 | 54 | 48 | 97 | 94 | 76 | 58 | 90 | 90 | 79 | 79 | 79 | 73 | 99 | 98 | 85 | 90 | 99 | 91 | 86 | 89 | 83 | 77 | 100 | 97 | 82 | 96 |
| Mult + IDXMult + CHM + Text | 83 | 88 | 57 | 77 | 56 | 70 | 80 | 80 | 60 | 36 | 74 | 81 | 77 | 79 | 62 | 78 | 92 | 96 | 68 | 46 | 89 | 94 | 99 | 81 | 79 | 93 | 92 | 96 | 76 | 73 |
| Mult + IDXMult + CHM + Stru | 72 | 85 | 60 | 72 | 59 | 57 | 54 | 65 | 67 | 46 | 69 | 69 | 73 | 59 | 59 | 71 | 70 | 68 | 73 | 71 | 97 | 80 | 73 | 73 | 69 | 71 | 67 | 74 | 67 | 80 |
| Mult + Text + Stru | 79 | 65 | 79 | 93 | 56 | 68 | 86 | 80 | 56 | 54 | 76 | 60 | 91 | 98 | 67 | 69 | 79 | 98 | 56 | 54 | 94 | 78 | 99 | 94 | 85 | 77 | 89 | 96 | 56 | 88 |
| PCA | 89 | 89 | 61 | 77 | 72 | 58 | 90 | 87 | 71 | 73 | 94 | 94 | 71 | 74 | 72 | 64 | 93 | 90 | 68 | 75 | 97 | 92 | 82 | 77 | 80 | 69 | 93 | 97 | 65 | 83 |
| **Dry Season/Layer** | | | | | | | | | | | | | | | | | | | | | | | | | | | | | | |
| RGB + IDXRGB + CHM | 56 | 64 | 74 | 77 | 51 | 66 | 87 | 84 | 60 | 42 | 60 | 63 | 90 | 85 | 70 | 81 | 93 | 93 | 64 | 55 | 72 | 75 | 94 | 85 | 78 | 85 | 97 | 94 | 76 | 76 |
| RGB + IDXRGB + CHM + text | 61 | 82 | 74 | 80 | 54 | 63 | 84 | 78 | 68 | 44 | 64 | 89 | 82 | 76 | 71 | 63 | 80 | 77 | 72 | 67 | 81 | 83 | 93 | 76 | 83 | 73 | 80 | 95 | 68 | 90 |
| RGB + IDXRGB + CHM + stru | 81 | 81 | 48 | 79 | 57 | 52 | 81 | 76 | 67 | 53 | 94 | 69 | 71 | 71 | 61 | 68 | 78 | 80 | 48 | 65 | 87 | 84 | 81 | 86 | 89 | 60 | 87 | 96 | 59 | 94 |
| Mult + IDXMult + CHM + Text | 64 | 67 | 59 | 76 | 53 | 63 | 81 | 68 | 68 | 53 | 79 | 61 | 66 | 78 | 56 | 58 | 84 | 82 | 64 | 73 | 96 | 75 | 78 | 83 | 50 | 84 | 97 | 100 | 92 | 70 |
| Mult + IDXMult + CHM + Stru | 60 | 78 | 76 | 67 | 52 | 59 | 61 | 61 | 67 | 50 | 71 | 74 | 67 | 50 | 52 | 53 | 39 | 46 | 59 | 59 | 77 | 84 | 71 | 65 | 61 | 59 | 64 | 66 | 67 | 64 |
| Mult + Text + Stru | 78 | 78 | 64 | 81 | 69 | 61 | 93 | 93 | 77 | 67 | 74 | 74 | 77 | 71 | 41 | 50 | 96 | 100 | 89 | 79 | 85 | 82 | 67 | 76 | 66 | 59 | 100 | 97 | 81 | 81 |
| PCA | 71 | 73 | 65 | 79 | 59 | 80 | 88 | 65 | 72 | 66 | 74 | 70 | 74 | 85 | 82 | 74 | 81 | 81 | 69 | 77 | 87 | 79 | 83 | 79 | 85 | 76 | 84 | 93 | 79 | 96 |

**Table 4.** Confusion matrices involving RGB + IDXRGB + CHM, PCA, Mult + Text + Stru, RGB + IDXRGB + CHM input layers, random forest (RF), decision tree (DT), and support vector machine (SVM) classifiers for wet and dry seasons. T.P—*Trembleya parviflora*; B.Rw—*Baccharis retusa* (wood); B.Rg—*Baccharis retusa* (green Canopy), Shadow (Sh), and Ot—Others. For abbreviation identification of input layers, see Table 1.

| | Wet Season | | | | | | Dry Season | | | | |
|---|---|---|---|---|---|---|---|---|---|---|---|
| | RGB + IDXRGB + CHM (RF) | | | | | | PCA (RF) | | | | |
| | T.P | B.Rw | B.Rg | Sh | Ot | | T.P | B.Rw | B.Rg | Sh | Ot |
| T.P | 24 | 0 | 0 | 0 | 0 | T.P | 27 | 0 | 2 | 2 | 0 |
| B.Rw | 0 | 30 | 0 | 1 | 0 | B.Rw | 0 | 19 | 4 | 0 | 0 |
| B.Rg | 3 | 3 | 26 | 0 | 0 | B.Rg | 2 | 2 | 29 | 0 | 1 |
| Sh | 0 | 0 | 0 | 35 | 0 | Sh | 2 | 0 | 3 | 27 | 0 |
| Ot | 3 | 1 | 0 | 0 | 24 | Ot | 3 | 3 | 0 | 0 | 23 |
| | Mult + Text + Stru (DT) | | | | | | Mult + Text + Stru (DT) | | | | |
| | T.P | B.Rw | B.Rg | Sh | Ot | | T.P | B.Rw | B.Rg | Sh | Ot |
| T.P | 25 | 0 | 3 | 0 | 5 | T.P | 20 | 0 | 5 | 0 | 2 |
| B.Rw | 0 | 31 | 2 | 0 | 1 | B.Rw | 0 | 30 | 7 | 0 | 2 |
| B.Rg | 3 | 0 | 18 | 0 | 6 | B.Rg | 4 | 11 | 12 | 0 | 2 |
| Sh | 5 | 0 | 0 | 22 | 1 | Sh | 1 | 0 | 0 | 27 | 0 |
| Ot | 9 | 0 | 3 | 0 | 15 | Ot | 2 | 1 | 0 | 0 | 23 |
| | PCA (SVM) | | | | | | RGB + IDXRGB + CHM (SVM) | | | | |
| | T.P | B.Rw | B.Rg | Sh | Ot | | T.P | B.Rw | B.Rg | Sh | Ot |
| T.P | 32 | 0 | 1 | 2 | 1 | T.P | 14 | 0 | 2 | 2 | 7 |
| B.Rw | 1 | 17 | 6 | 0 | 4 | B.Rw | 0 | 23 | 5 | 1 | 2 |
| B.Rg | 0 | 3 | 18 | 1 | 3 | B.Rg | 3 | 3 | 19 | 1 | 11 |
| Sh | 1 | 0 | 2 | 27 | 0 | Sh | 1 | 2 | 0 | 26 | 1 |
| Ot | 2 | 2 | 4 | 1 | 22 | Ot | 4 | 2 | 3 | 1 | 15 |

Figure 3 presents a sample of the SVM, DT, and RF classifications. We chose transect 2 because it shows a clear transition of *B. retusa* encroachment on dry grassland (campo sujo) to a *T. parviflora* encroachment on wet grassland (campo úmido). Figure 3 also shows all land cover classes other than Sh. On the left, we can see a mixture of the two *B. retusa* classes but with a predominance of the B.Rg class. Near the center of the images, (a) and (b), B.Rw becomes predominant, whereas, in the southwest portion of the dry season image, an extensive canopy of a third species representing the others (Ot) class can be clearly observed. The six classification images were chosen to broadly represent the 42 resultant classification maps.

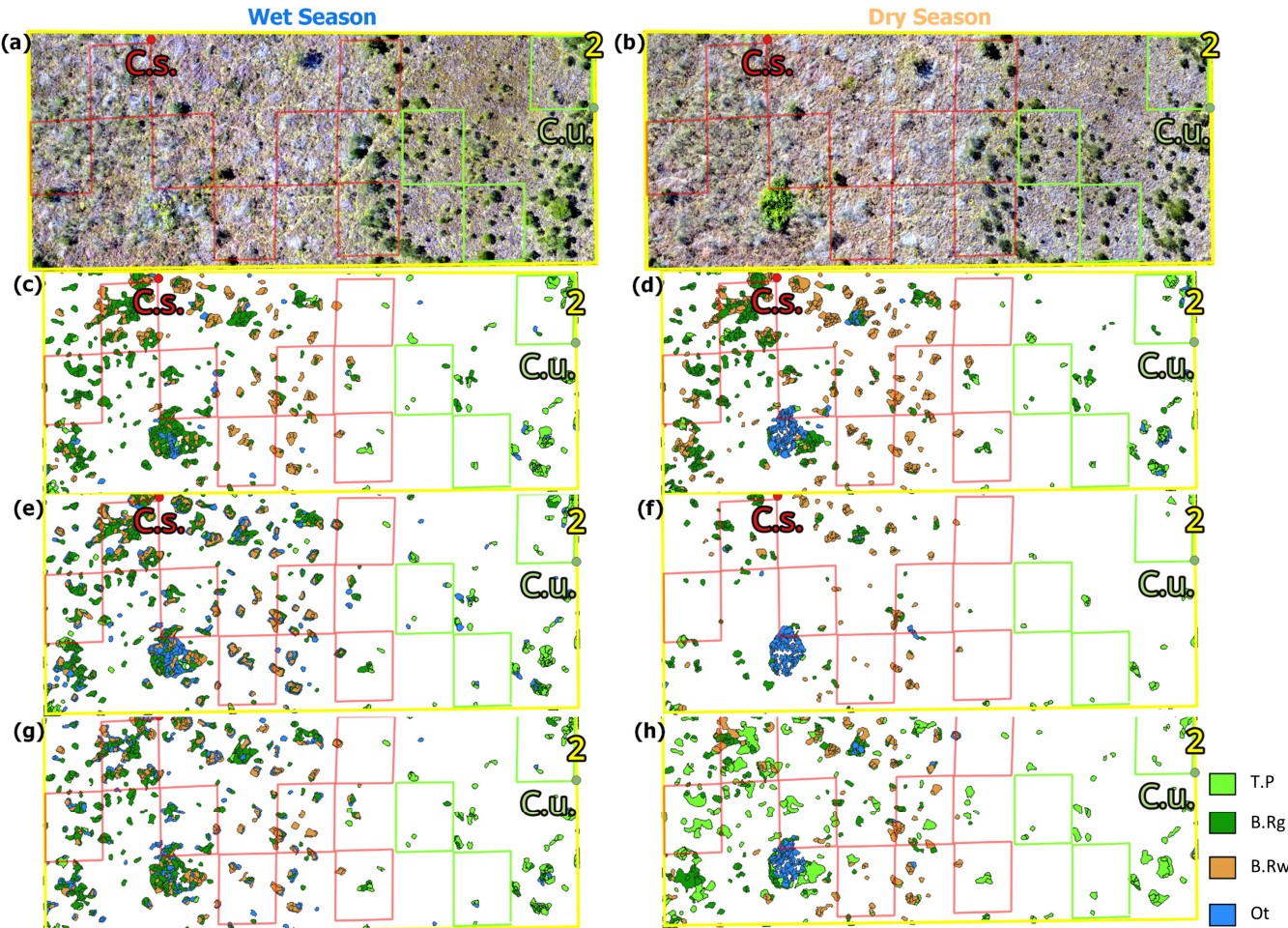

**Figure 3.** Six among 42 woody plant encroachment classification maps from the transect 2. Green and red quadrants are campo úmido (wet grasslands) and campo sujo (grasslands with scattered woody plants), respectively. The 10 cm pixel size, RGB reflectance orthomosaics acquired during the wet and dry seasons are shown in (**a**,**b**), respectively; (**c**) The resulting classification of RGB + IDXRGB + CHM input layer and random forest (RF) classifier for the wet season; (**d**) the resulting classification of PCA input layer and RF classifier for the dry season; (**e**,**f**) the resulting classification of Mult + Text + Stru input layer and decision tree (DT) classifier for the wet and dry seasons, respectively; (**g**) the resulting classification of PCA input layer and support vector machine (SVM) classifier for the wet season; (**h**) the resulting classification of RGB + IDXRGB + CHM input layer and SVM classifier for the dry season. T.P = *Trembleya parviflora*; B.Rw = *Baccharis retusa* (wood); B.Rg = *Baccharis retusa* (green canopy); and Ot = Others. For abbreviation identification of input layers, see Table 1.

In order to assess the validity of the classification models, Figure 4 presents nine learning curves generated from the models selected to extract the subset data for in situ validation. All models (classifier + input layer) were considered valid since the training and cross-validation score curves decrease their distances according to the increase in training size. The tipping point is approximately 150 samples for all DT and RF models, while for the SVM models, this point corresponds to approximately 350 samples.

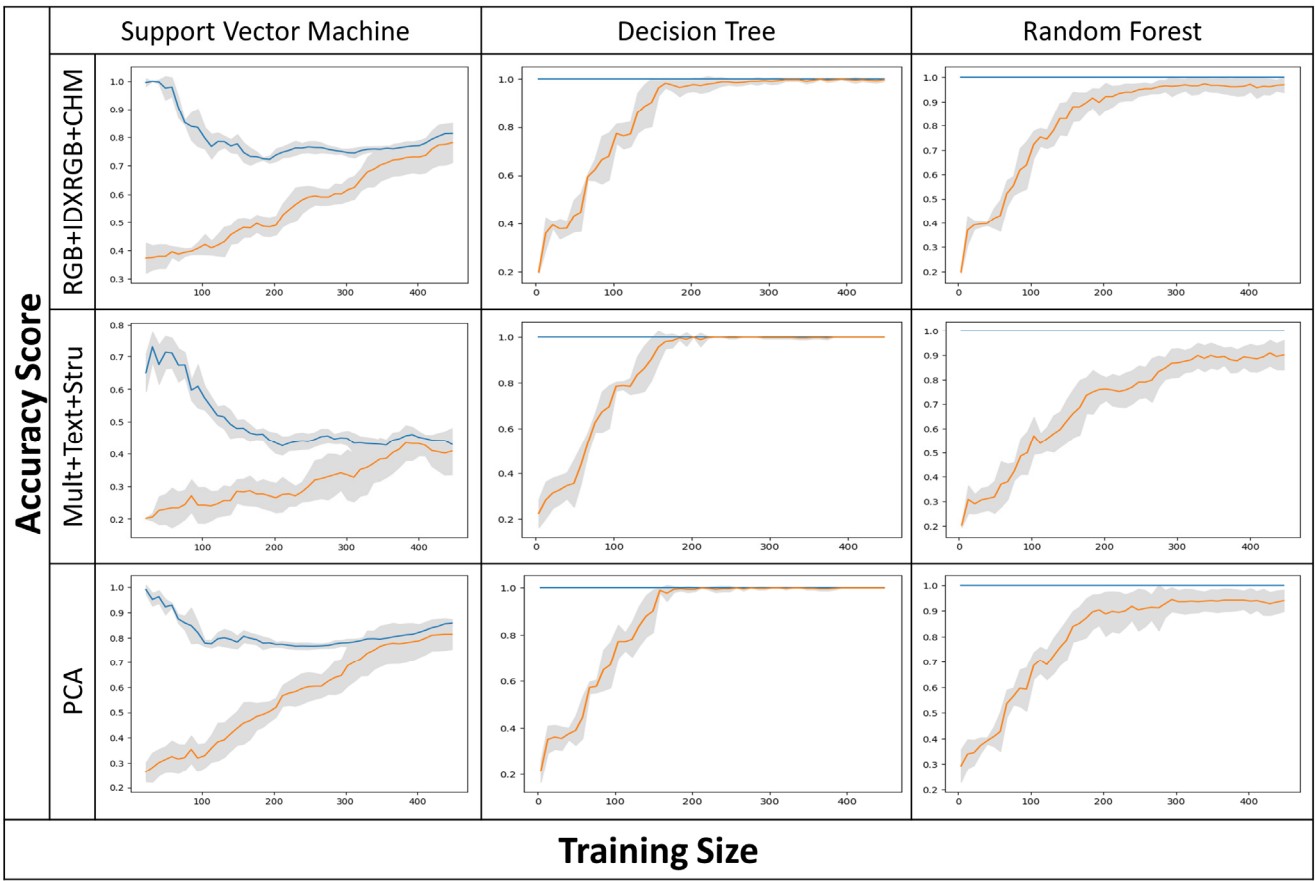

**Figure 4.** Learning curves relating to classification models: support vector machine (SVM), decision tree (DT), and random forest (RF), and input layers from wet season orthomosaic. The blue and orange lines with their standard deviation (SD) are the training and cross-validation mean scores, respectively. For abbreviation identification of input layers, see Table 1.

In addition to the validity of the selected models, Table 5 presents three performance indicators (Precision, Recall, and F-score) of the models for each of the four classes. Among the classes of interest, the T.P presented, on average, the highest values of the indicators, followed by the B.Rw and B.Rg. With regards to the classifiers, the pattern was repeated, and RF led followed by DT and SVM. Finally, the input layer with the highest average indicators was RGB + IDXRGB + CHM followed by PCA.

### 3.2. In Situ Validation and Cerrado Phytophysiognomy Accuracy Assessment

A comparison of the two forms of validation, in situ and 30% of the data, showed that average OA and Kappa coefficients for in situ validation were lower for classifiers and input layers (Table 6). Despite the model with the highest OA (85%) being DT with the PCA input layer, the RF showed the smallest variation among all predictors. Yet, there was no statistical difference among the classifiers' performances according to the McNemar test. Unlike validation using 30% of the data, the input layer with the highest average OA and Kappa coefficient was PCA (Table 6). The level of agreement was relatively high for all layers and classifiers, with a Kappa coefficient ranging from 0.62 to 0.81.

**Table 5.** Precision (P), Recall (R), and F-score (F) are displayed for three selected input layers, and three machine learning classifiers. T.P—*Trembleya parviflora*; B.Rw—*Baccharis retusa* (wood); B.Rg—*Baccharis retusa* (green canopy), Sh—Shadow and Ot—Others. For abbreviation identification of input layers, see Table 1.

| Classifier | Support Vector Machine | | | | | | | | | | | | | | |
|---|---|---|---|---|---|---|---|---|---|---|---|---|---|---|---|
| Class | T.P | | | B.Rw | | | B.Rg | | | Sh | | | Ot | | |
| Indicators | P | R | F | P | R | F | P | R | F | P | R | F | P | R | F |
| RGB + IDXRGB + CHM | 0.78 | 0.87 | 0.82 | 0.92 | 0.80 | 0.86 | 0.73 | 0.68 | 0.71 | 0.94 | 1.00 | 0.97 | 0.76 | 0.78 | 0.77 |
| Mult + Text + Stru | 0.65 | 0.79 | 0.72 | 0.93 | 0.79 | 0.86 | 0.68 | 0.56 | 0.61 | 0.80 | 0.86 | 0.83 | 0.53 | 0.56 | 0.54 |
| PCA | 0.89 | 0.89 | 0.89 | 0.77 | 0.61 | 0.68 | 0.58 | 0.72 | 0.64 | 0.87 | 0.90 | 0.88 | 0.73 | 0.71 | 0.72 |
| Classifier | Decision Tree | | | | | | | | | | | | | | |
| Class | T.P | | | B.Rw | | | B.Rg | | | Sh | | | Ot | | |
| Indicators | P | R | F | P | R | F | P | R | F | P | R | F | P | R | F |
| RGB + IDXRGB + CHM | 0.82 | 0.96 | 0.88 | 0.79 | 0.87 | 0.83 | 0.87 | 0.81 | 0.84 | 0.97 | 0.97 | 0.97 | 0.87 | 0.71 | 0.78 |
| Mult + Text + Stru | 0.59 | 0.76 | 0.67 | 1.00 | 0.92 | 0.95 | 0.69 | 0.67 | 0.68 | 1.00 | 0.78 | 0.88 | 0.53 | 0.56 | 0.55 |
| PCA | 0.94 | 0.94 | 0.94 | 0.74 | 0.71 | 0.72 | 0.64 | 0.72 | 0.68 | 0.90 | 0.93 | 0.92 | 0.75 | 0.68 | 0.72 |
| Classifier | Random Forest | | | | | | | | | | | | | | |
| Class | T.P | | | B.Rw | | | B.Rg | | | Sh | | | Ot | | |
| Indicators | P | R | F | P | R | F | P | R | F | P | R | F | P | R | F |
| RGB + IDXRGB + CHM | 0.80 | 1.00 | 0.89 | 0.88 | 0.97 | 0.92 | 1.00 | 0.81 | 0.90 | 0.97 | 1.00 | 0.98 | 1.00 | 0.86 | 0.92 |
| Mult + Text + Stru | 0.77 | 0.93 | 0.85 | 0.94 | 1.00 | 0.97 | 0.77 | 0.85 | 0.81 | 0.96 | 0.90 | 0.92 | 0.88 | 0.56 | 0.69 |
| PCA | 0.92 | 0.97 | 0.94 | 0.76 | 0.82 | 0.79 | 0.69 | 0.80 | 0.74 | 0.96 | 0.93 | 0.95 | 0.83 | 0.64 | 0.72 |

**Table 6.** Overall accuracies (AOs) and Kappa coefficients for three input layers and three machine learning classifiers from in situ reference data. For abbreviation identification of input layers, see Table 1.

| Layers/Classifier | SVM | DT | RF |
|---|---|---|---|
| RGB + IDXRGB + CHM | 78.4 (0.71) | 78.3 (0.71) | 75.6 (0.67) |
| Mult + Text + Stru | 72.1 (0.63) | 71.4 (0.62) | 72.3 (0.64) |
| PCA | 76.7 (0.69) | 85.4 (0.81) | 81.8 (0.76) |

Figure 5 shows the learning curves generated from the subset of selected data for in situ validation. This subset represents about 10% of all data, and apparently, it is necessary to increase the total training size since the training and cross-validation score curves are not close enough. Only models that combine the PCA input layer with the DT and SVM classifiers show an acceptable pattern. The other learning curves show patterns similar to those of underfitting models.

In terms of accuracy, commission and omission errors were generally smallest for B.Rw, followed by B.Rg, and were largest for *T. parviflora* (Tables 7, 8 and S2). Evaluation of all classifiers and input layers showed that the producer's accuracy (PA) was generally higher than the user's accuracy (UA); that is, there were fewer errors of omission than of commission. Regarding input layers, accuracies were generally highest for PCA followed by RGB + IDXRGB + CHM. However, UA was generally higher for the Mult + Text + Stru input layer than the RGB input layer. Finally, commission and omission errors were lowest for the DT classifier, and the average PA for the RF classifier was nearly the same as that of DT. The SVM was the only classifier for which commission errors were lower than omission errors (Tables 7, 8 and S2).

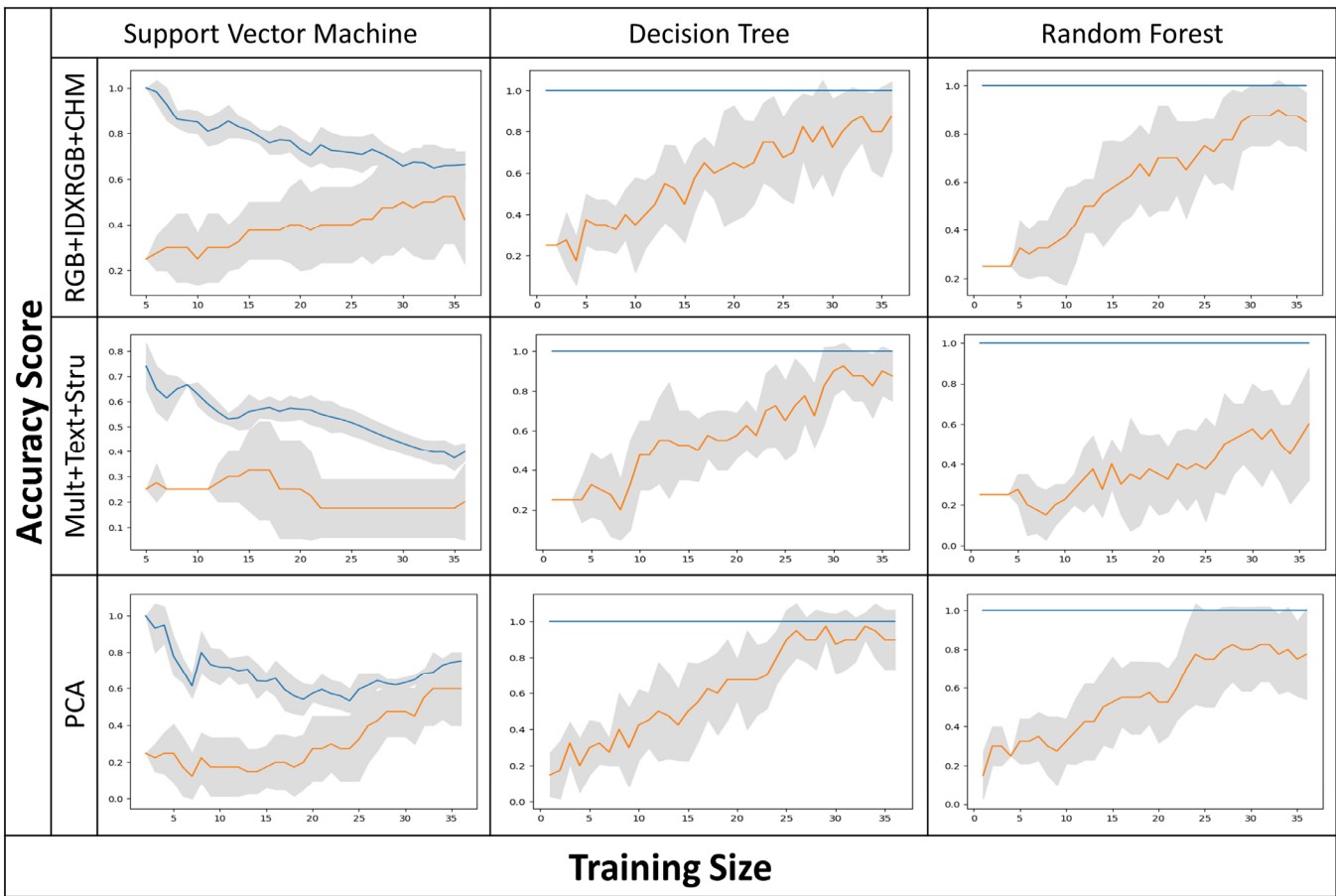

**Figure 5.** Learning curves relating to classification models: support vector machine (SVM), decision tree (DT), and random forest (RF), and input layers from subset for in situ validation of wet season orthomosaic. The blue and orange lines with their standard deviation (SD) are the training and cross-validation mean scores, respectively. For abbreviation identification of input layers, see Table 1.

Regarding accuracy assessments for the Cerrado phytophysiognomies, in general, the average OA was highest for campos úmidos (wet grasslands) at 78.9%, followed by cerrados ralos (open savannas), campos sujos (grasslands with scattered woody plants), and cerrados típicos (typical savannas) at 71.5% (Table 9). Notably, the accuracy of the classifiers depended on the phytophysiognomy. For example, the average OA of the RF classifier was highest on wet grasslands (80.2%), whereas SVM was the most accurate classifier for cerrado típico, and DT was the most accurate classifier for campo sujo (80.0%). Regarding the input layers, the average OA was highest for PCA, specifically for the grasslands: campo sujo (83.9%) and campo úmido, (83.3%). The second best-performing input layer was RGB + IDX + CHM with an average of 75.19% OA, followed by Mult + Text + Stru (72.61% of OA).

**Table 7.** User's (UA) and producer's accuracy (PA) of each class of land cover: Trembleya parviflora (T.P.), Baccharis retusa—wood (B.Rw), Baccharis retusa—green canopy (B.Rg), and Others (Ot). For abbreviation identification of input layers, see Table 1.

| Classifier | Support Vector Machine | | | | | | | | Decision Tree | | | | | | | | Random Forest | | | | | | | |
|---|---|---|---|---|---|---|---|---|---|---|---|---|---|---|---|---|---|---|---|---|---|---|---|---|
| Class | T.P | | B.Rw | | B.Rg | | Ot | | T.P | | B.Rw | | B.Rg | | Ot | | T.P | | B.Rw | | B.Rg | | Ot | |
| Accuracy | UA | PA | UA | PA | UA | PA | UA | PA | UA | PA | UA | PA | UA | PA | UA | PA | UA | PA | UA | PA | UA | PA | UA | PA |
| RGB + IDXRGB + CHM | 40.0 | 100.0 | 81.8 | 100.0 | 100.0 | 100.0 | 100.0 | 52.9 | 46.2 | 100.0 | 83.3 | 90.9 | 100.0 | 100.0 | 88.9 | 47.1 | 46.2 | 100.0 | 83.3 | 90.9 | 100.0 | 100.0 | 88.9 | 47.1 |
| Mult + Text + Stru | 38.5 | 83.3 | 100.0 | 100.0 | 80.0 | 88.9 | 77.8 | 41.2 | 28.6 | 80.0 | 87.5 | 100.0 | 100.0 | 100.0 | 90.0 | 45.0 | 28.6 | 80.0 | 87.5 | 100.0 | 100.0 | 100.0 | 90.0 | 45.0 |
| PCA | 70.0 | 100.0 | 72.7 | 100.0 | 70.0 | 100.0 | 100.0 | 47.4 | 76.9 | 100.0 | 88.9 | 100.0 | 90.0 | 90.0 | 88.9 | 61.5 | 76.9 | 100.0 | 88.9 | 100.0 | 90.0 | 90.0 | 88.9 | 61.5 |

**Table 8.** Confusion matrices involving RGB + IDXRGB + CHM, Mult + Text + Stru, and PCA input layer, support vector machine (SVM), decision tree (DT), and random forest (RF) classifiers from in situ reference data. T.P—*Trembleya parviflora*; B.Rw—*Baccharis retusa* (wood); B.Rg—*Baccharis retusa* (green canopy), and Ot—Others. For abbreviation identification of input layers, see Table 1.

| Support Vector Machine | | | | | Decision Tree | | | | | Random Forest | | | |
|---|---|---|---|---|---|---|---|---|---|---|---|---|---|
| **RGB + IDXRGB + CHM** | | | | | **RGB + IDXRGB + CHM** | | | | | **RGB + IDXRGB + CHM** | | | |
| | T.P | B.Rw | B.Rg | Ot | | T.P | B.Rw | B.Rg | Ot | | T.P | B.Rw | B.Rg | Ot |
| T.P | 4 | 0 | 0 | 6 | T.P | 6 | 0 | 0 | 7 | T.P | 6 | 0 | 0 | 7 |
| B.Rw | 1 | 10 | 0 | 1 | B.Rw | 0 | 12 | 0 | 0 | B.Rw | 0 | 7 | 0 | 3 |
| B.Rg | 0 | 0 | 13 | 2 | B.Rg | 0 | 0 | 10 | 2 | B.Rg | 0 | 0 | 7 | 0 |
| Ot | 4 | 0 | 0 | 12 | Ot | 0 | 0 | 1 | 8 | Ot | 0 | 0 | 0 | 11 |
| **Mult + Text + Stru** | | | | | **Mult + Text + Stru** | | | | | **Mult + Text + Stru** | | | |
| | T.P | B.Rw | B.Rg | Ot | | T.P | B.Rw | B.Rg | Ot | | T.P | B.Rw | B.Rg | Ot |
| T.P | 5 | 0 | 0 | 8 | T.P | 4 | 0 | 0 | 10 | T.P | 8 | 0 | 0 | 8 |
| B.Rw | 0 | 11 | 0 | 2 | B.Rw | 0 | 10 | 0 | 0 | B.Rw | 0 | 9 | 0 | 1 |
| B.Rg | 0 | 0 | 10 | 0 | B.Rg | 0 | 0 | 7 | 1 | B.Rg | 0 | 0 | 9 | 3 |
| Ot | 4 | 1 | 0 | 8 | Ot | 1 | 0 | 0 | 9 | Ot | 0 | 0 | 1 | 8 |
| **PCA** | | | | | **PCA** | | | | | **PCA** | | | |
| | T.P | B.Rw | B.Rg | Ot | | T.P | B.Rw | B.Rg | Ot | | T.P | B.Rw | B.Rg | Ot |
| T.P | 7 | 0 | 0 | 3 | T.P | 10 | 0 | 0 | 3 | T.P | 8 | 0 | 0 | 4 |
| B.Rw | 0 | 8 | 0 | 1 | B.Rw | 0 | 9 | 0 | 1 | B.Rw | 0 | 11 | 0 | 2 |
| B.Rg | 0 | 0 | 10 | 3 | B.Rg | 0 | 0 | 8 | 1 | B.Rg | 0 | 0 | 9 | 1 |
| Ot | 4 | 0 | 0 | 8 | Ot | 0 | 1 | 0 | 8 | Ot | 1 | 0 | 0 | 8 |

**Table 9.** Overall accuracies (AOs) and the Kappa coefficients or three input layers and three machine learning classifiers generated from in situ reference data from four Cerrado's phytophysiognomies. For abbreviation identification of input layers, see Table 1.

| Layers | Grasslands | | | | | | Woodlands | | | | | |
|---|---|---|---|---|---|---|---|---|---|---|---|---|
| | Campo Úmido | | | Campo Sujo | | | Cerrado Ralo | | | Cerrado Típico | | |
| RGB + IDXRGB + CHM | 66.7 (0.53) | 80.0 (0.74) | 77.8 (0.70) | 75.0 (0.67) | 83.3 (0.76) | 77.8 (0.71) | 75.0 (0.65) | 83.3 (0.76) | 75.0 (0.62) | 83.3 (0.77) | 58.3 (0.48) | 66.7 (0.55) |
| Mult + Text + Stru | 87.5 (0.83) | 75.0 (0.65) | 72.7 (0.58) | 72.7 (0.63) | 66.7 (0.56) | 75.0 (0.64) | 66.7 (0.56) | 80.0 (0.71) | 83.3 (0.78) | 66.7 (0.56) | 66.7 (0.53) | 58.3 (0.46) |
| PCA | 70.0 (0.59) | 90.0 (0.83) | 90.0 (0.84) | 88.9 (0.81) | 90.0 (0.86) | 72.7 (0.64) | 83.3 (0.76) | 75.0 (0.65) | 83.3 (0.77) | 72.7 (0.61) | 88.9 (0.84) | 81.8 (0.76) |
| Classifiers | SVM | DT | RF | SVM | DT | RF | SVM | DT | RF | SVM | DT | RF |

In all classes, except Ot, the user's accuracy was lower than the producer's; that is, there was more commission than omission error; as for the Ot class, the pattern was the opposite (Tables 10 and S2). The class with the highest user and producer accuracies was B.Rw followed by B.Rg, T.P, and finally, Ot. The campo úmido had the lowest mean commission and omission errors.

**Table 10.** User's (UA) and producer's accuracy (PA) of each class of land cover: Trembleya parviflora (T.P.), Baccharis retusa—wood (B.Rw), Baccharis retusa—green canopy (B.Rg), and others (Ot) for three input layers and three machine learning classifiers generated from in situ reference data for the four Cerrado phytophysiognomies. For abbreviation identification of input layers, see Table 1.

| Classifier | Support Vector Machine | | | | | | | | Decision Tree | | | | | | | | Random Forest | | | | | | | |
|---|---|---|---|---|---|---|---|---|---|---|---|---|---|---|---|---|---|---|---|---|---|---|---|---|
| Class | T.P | | B.Rw | | B.Rg | | Ot | | T.P | | B.Rw | | B.Rg | | Ot | | T.P | | B.Rw | | B.Rg | | Ot | |
| Campo Úmido | UA | PA | UA | PA | UA | PA | UA | PA | UA | PA | UA | PA | UA | PA | UA | PA | UA | PA | UA | PA | UA | PA | UA | PA |
| RGB + IDXRGB + CHM | 66.7 | 50.0 | 100.0 | 100.0 | 50.0 | 100.0 | 50.0 | 50.0 | 100.0 | 100.0 | 33.3 | 100.0 | 100.0 | 100.0 | 100.0 | 50.0 | 100.0 | 66.7 | 100.0 | 100.0 | 50.0 | 100.0 | 66.7 | 66.7 |
| Mult + Text + Stru | 100.0 | 66.7 | 100.0 | 100.0 | 100.0 | 100.0 | 50.0 | 100.0 | 50.0 | 50.0 | 100.0 | 100.0 | 100.0 | 100.0 | 66.7 | 100.0 | 71.4 | 100.0 | 50.0 | 100.0 | 100.0 | 100.0 | 100.0 | 25.0 |
| PCA | 75.0 | 100.0 | 50.0 | 100.0 | 50.0 | 100.0 | 100.0 | 40.0 | 100.0 | 100.0 | 100.0 | 100.0 | 100.0 | 50.0 | 50.0 | 100.0 | 100.0 | 83.3 | 100.0 | 100.0 | 100.0 | 100.0 | 50.0 | 100.0 |
| **Campo Sujo** | | | | | | | | | | | | | | | | | | | | | | | | |
| RGB + IDXRGB + CHM | 33.3 | 100.0 | 100.0 | 100.0 | 100.0 | 100.0 | 100.0 | 50.0 | 50.0 | 100.0 | 100.0 | 83.3 | 100.0 | 100.0 | 66.7 | 66.7 | 33.3 | 100.0 | 100.0 | 100.0 | 100.0 | 100.0 | 100.0 | 33.3 |
| Mult + Text + Stru | 33.3 | 100.0 | 100.0 | 100.0 | 50.0 | 100.0 | 100.0 | 40.0 | 20.0 | 100.0 | 100.0 | 100.0 | 100.0 | 100.0 | 100.0 | 33.3 | 50.0 | 100.0 | 80.0 | 80.0 | 100.0 | 100.0 | 66.7 | 50.0 |
| PCA | 100.0 | 100.0 | 83.3 | 100.0 | 100.0 | 100.0 | 100.0 | 50.0 | 50.0 | 100.0 | 100.0 | 100.0 | 100.0 | 100.0 | 100.0 | 66.7 | 50.0 | 100.0 | 75.0 | 100.0 | 66.7 | 100.0 | 100.0 | 40.0 |
| **Cerrado Ralo** | | | | | | | | | | | | | | | | | | | | | | | | |
| RGB + IDXRGB + CHM | 100.0 | 80.0 | 100.0 | 100.0 | 100.0 | 100.0 | 50.0 | 100.0 | 50.0 | 100.0 | 100.0 | 100.0 | 83.3 | 100.0 | 100.0 | 50.0 | 33.3 | 100.0 | 100.0 | 100.0 | 66.7 | 100.0 | 100.0 | 62.5 |
| Mult + Text + Stru | 100.0 | 100.0 | 100.0 | 100.0 | 80.0 | 50.0 | 25.0 | 100.0 | 50.0 | 100.0 | 50.0 | 100.0 | 100.0 | 100.0 | 100.0 | 33.3 | 33.3 | 100.0 | 100.0 | 100.0 | 100.0 | 100.0 | 100.0 | 60.0 |
| PCA | 100.0 | 50.0 | 100.0 | 100.0 | 100.0 | 100.0 | 66.7 | 100.0 | 50.0 | 100.0 | 66.7 | 100.0 | 66.7 | 100.0 | 100.0 | 57.1 | 50.0 | 100.0 | 100.0 | 100.0 | 75.0 | 100.0 | 100.0 | 60.0 |
| **Cerrado Típico** | | | | | | | | | | | | | | | | | | | | | | | | |
| RGB + IDXRGB + CHM | 100.0 | 100.0 | 100.0 | 80.0 | 100.0 | 100.0 | 60.0 | 100.0 | 16.7 | 100.0 | 100.0 | 100.0 | 100.0 | 100.0 | 100.0 | 28.6 | 40.0 | 100.0 | 100.0 | 100.0 | 66.7 | 100.0 | 100.0 | 42.9 |
| Mult + Text + Stru | 100.0 | 100.0 | 100.0 | 66.7 | 100.0 | 100.0 | 42.9 | 100.0 | 20.0 | 100.0 | 100.0 | 100.0 | 100.0 | 100.0 | 100.0 | 50.0 | 25.0 | 100.0 | 50.0 | 100.0 | 75.0 | 100.0 | 100.0 | 28.6 |
| PCA | 100.0 | 100.0 | 100.0 | 50.0 | 100.0 | 100.0 | 40.0 | 100.0 | 66.7 | 100.0 | 100.0 | 100.0 | 100.0 | 100.0 | 100.0 | 50.0 | 33.3 | 100.0 | 100.0 | 100.0 | 100.0 | 100.0 | 100.0 | 50.0 |

## 4. Discussion

### 4.1. Overall Accuracy Assessment and Model Validity

Our analysis of all combinations of input layers, classifiers, and seasons showed that the highest level of agreement was near perfect (Kappa coefficient = 0.91), and the lowest was moderate (Kappa coefficient = 0.46). These results demonstrate that classifications were acceptable even using the worst models tested and confirmed the feasibility of using UAV images to classify woody encroachment in tropical savannas. Validating the classifiers using 30% of the samples resulted in the best map generated by the wet season orthomosaic using the RGB sensor and RF classifier. Our results agree with those of Olariu et al. [52], who reported satisfactory woody encroachment classification using only the RGB sensor in a semi-arid region. However, this result only partially corroborates our hypothesis. We noted that the segmentation vectors directly influenced map accuracy. Segments that were closer to the entire canopy (i.e., not oversegmented) resulted in more accurate maps. Multispectral sensors captured more canopy details, resulting in smaller segments, which decreased the accuracy of the multispectral maps and increased processing time.

The most accurate classifications were obtained during the wet season, regardless of the input layer and classifier. The accuracy of classifying *B. retusa* (wood) was less influenced by season but this was still high, which was expected because the leafless wood canopy tends to maintain its spectral signature regardless of water availability. This is a positive result for managing and monitoring woody encroachment, especially in wet grasslands, because the wet season is the most suitable period for floristic surveys [23,82] and monitoring of water dynamics [83–85].

The accuracy of the RF classifier was the highest among the chosen algorithms. It is worth noting that by standardizing the same segments for training and validating the classifiers, it was possible to fairly compare the performance of each one. This is because classification accuracy is strongly influenced by the quality of the training and validation data. The nature of these data can have an even greater impact than the algorithm itself [86]. However, there are differences in the behavior of each of our classifiers, especially concerning the areas (e.g., pixel values) chosen to represent each class. In general, we selected areas to represent the typical range of values for each class. However, the SVM algorithm works by using a subset of the input layers to define the boundary or margin conditions of each class. Thus, the strategy to collect data for SVM should focus on the boundary values that differentiate the classes rather than the typical ranges of values [59]. This may explain our finding that SVM was the model with the lowest accuracy, although the proper data collection strategy for SVM is not clear [59].

Learning curves describe the performance of a process on a task as a function of some resource for solving that task. Here the task is an image classification and the resource is the training size. When analyzing learning curves for machine learning, there is a trade-off between bias and variance. It means the model must be good enough to represent input data specifics but at the same time simplify complex patterns. When we worked with all our data, it was possible to verify the validity of all our models from the shape of learning curves, an addition to highlighting the tipping point where the models tend to saturate. When we used the DT and RF, this tipping point occurred earlier than for the SVM models, with 150 and 350 training sizes, respectively (Figure 4).

### 4.2. In Situ Validation

In general, in situ validation produced less accurate classification maps than validation using 30% of the data. Although validation using 30% of the data resulted in classifications with moderate to a near-perfect agreement, the in situ validation method produced Kappa coefficients that fluctuated considerably. Creating land cover maps with remote sensing images is not difficult; however, the accuracy of the resulting map will depend on data quality and classifier performance [87]. Analyzing the learning curves made from a subset of the data, it was possible to verify that the models lost performance, and it was necessary for an increase in the training size to overcome the underfitting problem.

Thus, it is advisable to create multiple maps using more than one input data source and multiple classification models for comparison [60] to achieve the map that best represents reality depending on the rigor applied in assessing map accuracy [88]. In a review of accuracy assessment for land cover mapping, Stehman and Foody [89] summarized the key issues as sampling design, response design, and analysis. Our result is important, especially to define the minimum sample number for accuracy assessment that is heavily resource-dependent (e.g., fieldwork), such as the in situ validation.

Sampling design influences data quality and can be defined as the rules chosen to select subsets of assessment units for which the reference classification is obtained and then compared with the map classification. Response design defines how the agreement between the predicted map class label and the reference class label is decided. Reference data for the response design can include fieldwork specifically undertaken for the accuracy assessment, aerial photographs, airborne video, and even fine spatial resolution satellite images. Finally, the analysis summarizes information to quantify accuracy and enable comparison of the resulting maps [89,90]. In this study, we compared two approaches to response design: using reference data from the orthomosaics for validation (30%) and using data from fieldwork carried out specifically to acquire in situ reference values. Both methods involved random sampling and making an error matrix.

The DT and RF performances were, on average, very close: 85.4% and 81.8%, respectively. However, when comparing the different input layers, DT accuracy varied by about 7%, whereas RF accuracy varied by only 3%; thus, the accuracy of the RF classifier was more stable. This similar performance of the two classifiers can be explained by the nature of the algorithms [59]. The DT and RF classifiers are both based on decision trees; the difference is that RF is formed by several decision trees, which may account for its greater stability regardless of the input layer used [58]. However, this unusual result must be explained by an underfitting scenario since the performance of the RF is affected by small data size. Figure 5 shows cross-validation score curves far from saturation, which means that we did not reach the best training size.

In both presented validation methods, it was possible to observe that a considerable part of the commission errors of *T. parviflora* come from the category Others. During the in situ validation, it was possible to observe that these commission errors were mostly linked to species that were spectrally similar to *T. parviflora*, for example, palm trees (*Syagrus sp.*) with bright green leaves and trees of the genus *Vochysia* that have leaves with a shade of green very similar to that of *T. parviflora*.

### 4.3. Evaluating In Situ Accuracy in Different Phytophysiognomies

This is the first study to classify woody encroachment in different Brazilian savanna phytophysiognomies. Based on our ground truth sampling design, it was possible to compare the quality of woody species classification in four of the main woodland and grassland phytophysiognomies: cerrado típico, cerrado ralo, campo sujo, and campo úmido. The cerrado sensu stricto is the Cerrado's most extensive formation, occupying approximately 65% of the Brazilian savanna [91]. The cerrado sensu stricto is subdivided into cerrado denso, cerrado típico, cerrado ralo, and cerrado rupestre [41]. We were able to evaluate the classification of woody encroachment in two of the most important and representative phytophysiognomies of the Brazilian savanna. In addition, we evaluated the classification of woody encroachment in two grassland vegetations, one of them (campo úmido) being extremely relevant for conserving the functioning and provisioning of one of the most essential ecosystem services of the Cerrado, the water supply [77].

Beyond the importance of each vegetation formation, these four phytophysiognomies can be considered a continuum of woody density, from the cerrado típico, which is characterized by a woody cover of 20% to 50% and an average height of 3 to 6 m, to the campo úmido, which is a predominantly grassland formation with occasional shrubs and the complete absence of trees [41].

In our study, tree cover appeared to be a determining factor in the quality of the classification maps, with the highest OA for the campo úmido, and the lowest OA for the cerrado típico. Further study of this relationship between tree cover and accuracy may be needed because the two intermediate formations, campo sujo and cerrado ralo, showed the opposite pattern. Nevertheless, the classification accuracy of grassland formations was greater than that of woodland formations. Finally, we suggest that future studies also investigate the relationship between classifiers and phytophysiognomies because each classifier showed its best performance in different vegetation formations. For the input layers, the average accuracy of PCA was the highest, followed by RGB + IDXRGB + CHM. This result was, at some point, unexpected because more complex input layers tend to produce more accurate maps. However, the RGB products showed the results were as satisfactory as the multispectral ones, reinforcing the need, whenever possible, to combine different sensors in the identification and management of woody plant encroachment.

## 5. Conclusions

Our results demonstrate that low-cost drone images can be used to produce an acceptable classification of woody encroachment in tropical savannas. For this purpose, we recommend acquiring drone images during the wet season and using a combination of different sensors, when possible. Based on our in situ validation, the input layer with the best accuracy combined the products of the RGB and multispectral cameras. However, if it is not possible to combine two sensors, we suggest using the RGB sensor. The metrics and indices derived from the RGB sensor provided woody encroachment maps with high accuracy. We strongly recommend using more than one form of validation, with a preference for collecting in situ reference data, especially for studies of woody encroachment. Finally, we suggest, based on the analysis of the learning curves, that in the case of using in situ accuracy assessment, there are a minimum of 40 samples per class.

Regarding the choice of classifiers, we recommend using different combinations of input layers and at least three classifiers. Special care is needed regarding the sampling design to acquire training data for each classifier and its operating features (e.g., SVM). Finally, we welcome the advance in the use of deep learning for tree–shrub identification. However, machine learning is still a viable option and has produced satisfactory results.

As this is the first study to classify woody encroachment in Brazilian savanna phytophysiognomies, there is more research to be conducted. For example, there is a need for a more in-depth investigation of the relationship between the extent of tree cover and the accuracy of woody encroachment classification. Although a comparison of grassland classification and woodland classification showed greater accuracy within formations with less tree cover, this pattern was not observed for intermediate formations, such as grasslands with scattered woody plants and shrublands. Another question is the best choice of classifier for specific phytophysiognomies. Our results indicated the superiority of models based on decision trees (single DT and RF); however, the SVM classifier produced superior results in denser formations such as cerrado típico.

Finally, we suggest the need for future studies to investigate the use of UAV data as an alternative to field sampling, especially in tropical savannas and woodland formations. These types of data will be paramount for upscaling approaches (e.g., for satellite scale), enabling cost-effective monitoring of woody encroachment in time series.

**Supplementary Materials:** The following supporting information can be downloaded at: https://www.mdpi.com/article/10.3390/rs15092342/s1, **S1**. Confusion matrix and results of each class of land cover: *Trembleya parviflora*, *Baccharis retusa*—wood, *Baccharis retusa*—green canopy, shadow, and others for two seasons (wet and dry) from three machine learning classifiers. RGB refers to red, green, and blue bands; IDXRGB refers to the green leaf index and green–red difference index; CHM refers to the canopy height model; Mult refers to multispectral bands; Text refers to texture metric; Stru refers to structural metric; IDXMult refers to normalized difference vegetation index (NDVI) and normalized difference red edge (NDRE) index; and PCA refers to the six principal components of all bands; **S2**. Confusion matrix and results of each class of land cover: *Trembleya parviflora* (Tremb.),

*Baccharis retusa*—wood (Bac.S), *Baccharis retusa*—green canopy (Bac.V), and others (Outros.). RGB refers to red, green, and blue bands; IDXRGB refers to the green leaf index and green–red difference index; CHM refers to the canopy height model; Mult refers to multispectral bands; Text refers to texture metric; Stru refers to structural metric; and PCA refers to the six principal components of all bands.

**Author Contributions:** Study design: L.S.C., E.E.S., M.M.d.C.B., M.E.F., C.B.R.M. and T.R.B.d.M.; drone data acquisition: J.V.S.C. and L.R.A.J.; fieldwork: L.S.C. and T.R.B.d.M.; drone image preprocessing: J.V.S.C. and L.R.A.J.; analysis: L.S.C.; resources: M.M.d.C.B., M.E.F. and C.B.R.M.; writing the original manuscript: L.S.C.; writing—review and editing: L.S.C., E.E.S., M.M.d.C.B., M.E.F., C.B.R.M., T.R.B.d.M., J.V.S.C. and L.R.A.J.; supervision: M.M.d.C.B.; funding acquisition: M.M.d.C.B. All authors have read and agreed to the published version of the manuscript.

**Funding:** This study was funded by the Brazilian Long-term Ecological Research Program—PELD/ CNPq (grant # 312137/2021-4) and Fundação de Apoio a Pesquisa do Distrito Federal—FAP-DF (grant # 00193-00000229/2021-21). This study was also partially financed by the Coordenação de Aperfeiçoamento de Pessoal de Nível Superior (CAPES)—Finance code 001. E.E.S., M.E.F., and M.M.D.C.B. are CNPq research fellows (grants # 303502/2019-3, 315699/2020-5, and 441463/2017-7, respectively).

**Data Availability Statement:** Data are available upon request to the corresponding author.

**Acknowledgments:** We would like to thank the Ecosystem Ecology Laboratory (UnB), the Image Processing and GIS Laboratory—LAPIG/Socio-Environmental Studies Institute (UFG), and the Botanic Department (UnB). We also acknowledge the Botanic Garden of Brasília for logistics support. Additionally, we would like to thank the three reviewers for their helpful contributions.

**Conflicts of Interest:** The authors declare that they have no conflict of interest.

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
