# Peer review of "Woody Plant Encroachment in a Seasonal Tropical Savanna: Lessons about Classifiers and Accuracy from UAV Images"

_remotesensing, doi:10.3390/rs15092342_

Round 1

Reviewer 1 Report (Previous Reviewer 3)

I agree with your observations

Author Response

Thank you,

Best Regards

Reviewer 2 Report (Previous Reviewer 2)

Dear Authors,

I highly appreciate your effort incorporating comments. However, I'm not sure, if you have included or understood my comments to my earlier comments. You have tried to modify the input the machine learning models.

Please understand, the machine learning model development has the following steps:

1. Data Ingestion/Acquistion - RGB, multispectral, etc.

2. Data Processing/Feature Engineering - Conversion of the data so that the model has enough inputs to make predictions

3. Model development - Choice of Models, parameter Tuning, etc.

4. Model Evaluation - which is what your paper is all about.

==> You haven't mentioned the libraries you are using to implement the models. Are you using sklearn (Python) or something for Matlab?

==> Are you implementing the method yourself? if so, how are you confident that your implementation is not error proof.

==> RF and DT are similar. RF implements multiple DTs to arrive at the prediction. So, the result you have provided saying that DT performed better than RF, seems to be like a overfit scenario, as this is very unusual. Or there may be some information leak.

==> If you have provided any learning curves, that would have made it clear that your model is a valid good model. However, that is not there.

==> Did you use vanilla SVM or did you use any kernel tricks (RBM, etc)? Please be explicit.

==> Also, you haven't provided any F1 scores, precision, recall values, AUC, etc. Accuracy, is not always a good indicator.

==> PCA itself will not improve the accuracy of the model, it is a pre-processing to deal with curse of dimensionality, etc.

If you are really comparing, you should have provided why you have justify the choice of PCA for preprocessing, as it is not the only method.

Unfortunately, you have put in more effort in describing how you acquired the data and made the initial features but not on the choice of pre-processing or how you trained the models, etc.

Please provide the learning curves... and format the model development process, as I mentioned earlier. While your expectation that UAV images during wet season might make a difference due the reflectances between the canopy but I don't completely agree on how you came to the conclusion.

Further, the text needs to be improved as well. There are many misspellings, and many sentences needs rephrasing. Please correct them as well.

Author Response

Dear Authors,

I highly appreciate your effort incorporating comments. However, I'm not sure, if you have included or understood my comments to my earlier comments. You have tried to modify the input the machine learning models.

Thank you again for your valuable time in reviewing and advising on our study, your comments and suggestions are very precious to us. I would like to point out that no modifications were made to the input data, actually, due to the question raised earlier, we revised the number of in situ validation samples that were relatively uneven.

Please understand, the machine learning model development has the following steps:

  1. Data Ingestion/Acquistion - RGB, multispectral, etc.
  2. Data Processing/Feature Engineering - Conversion of the data so that the model has enough inputs to make predictions
  3. Model development - Choice of Models, parameter Tuning, etc.
  4. Model Evaluation - which is what your paper is all about.

Thank you for elucidate the model development steps. We agree that our main goal is generate the best model, however we believe that acquiring images in different seasons (wet and dry) and from different sensors (RGB and Mult) are important evidences raised by our study and fits in the first step (Acquistion). As well as the derivation of different indices (e. g. texture, CHM, Structure) falls under data processing. Finally, the choice of different groups of models such as: a discriminative non-parametric classifier (SVM), a decision tree and an assemble one (RF) was also a relevant process within the study.

Yet we still rewrote some steps to reinforce the weight of the other contributions of the work in machine learning model development.

==> You haven't mentioned the libraries you are using to implement the models. Are you using sklearn (Python) or something for Matlab?

Thank you for the question, we used a free and open-source software packages for geospatial analysis called Orfeo ToolBox (OTB, 2018). Developed by the French Center National d’Etudes Spatiales (CNES Development Team, 2018). OTB can be operated either autonomously or through a second open-source software (QGIS), used as a graphical interface that enables analysis of data processing. The choice of this toolbox is justified in the study because the algorithms are available, thus making their use accessible to a larger number of end users.

OTB uses the C++ library, based on Insight toolkit (ITK). Bindings are developed for Python. Since late 2009, some modules are developed as processing plugins for QGIS. Modules for classification, segmentation, hill shading have provided. OTB algorithms are now available in QGIS through the processing framework Sextante.

We understand that to prevent future doubts we have added the information in Material and Methods.

==> Are you implementing the method yourself? if so, how are you confident that your implementation is not error proof.

We understand and agree with your concern, however, as mentioned above, we use a free and open-source software packages for geospatial developed by the French Center National d’Etudes Spatiales (CNES) called Orfeo ToolBox (Grizonnet et al. 2017). This package has already been used by several studies (De Luca et al. 2019; Teodoro et al. 2016; Grizonnet etl al 2017; Cresson et al. 2018) which together have more than 400 citations. Based on this information, we consider that the OTB implementation is reliable.

Since there was a misunderstanding, we chose to add more details in the Methods section regarding the package used.

==> RF and DT are similar. RF implements multiple DTs to arrive at the prediction. So, the result you have provided saying that DT performed better than RF, seems to be like a overfit scenario, as this is very unusual. Or there may be some information leak.

Indeed, after analyzing the learning curve, we can observe that the limitation of 10 canopies per class, adding up to 40 statistical zones per combination of input and model, affected the model accuracies (Figure 5). Especially in RF there was an underfitting scenario. So we added the learning curves (Figure 4 and 5), the methods used to generate them, as well as the suggestion for future use of in situ validation method.

==> If you have provided any learning curves, that would have made it clear that your model is a valid good model. However, that is not there.

As suggested the learning curves have been added, see figures 4 and 5.

==> Did you use vanilla SVM or did you use any kernel tricks (RBM, etc)? Please be explicit.

Thanks for the suggestion and we added the information in lines 319-322.

==> Also, you haven't provided any F1 scores, precision, recall values, AUC, etc. Accuracy, is not always a good indicator.

We know that during the review process supplementary material is not accessed, so we have moved selected indicators: accuracy, recall and F1 scores to the main text in the table 5. We need to select only nine models as 42 were generated in all.

==> PCA itself will not improve the accuracy of the model, it is a pre-processing to deal with curse of dimensionality, etc.

The PCA was used as a dimensionality reduction tool, since large numbers of input features can cause poor performance for machine learning algorithms (Murphy. 2012), we believe that dimensionality reduction procedures may indirectly affect the accuracy of the model.

If you are really comparing, you should have provided why you have justify the choice of PCA for preprocessing, as it is not the only method.

PCA is the most common approach to dimensionality reduction (Murphy. 2012). Since we are targeting a larger number of end users we have chosen a technique of dimensionality reduction that is widely used and will be available in largely open-source software packages.

We appreciate the suggestion and have added this explanation to the main text.

Unfortunately, you have put in more effort in describing how you acquired the data and made the initial features but not on the choice of pre-processing or how you trained the models, etc.

Please provide the learning curves... and format the model development process, as I mentioned earlier.

We added the analysis of learning curves, the results are presented in the figures 4 and 5, as well as the methodology used and subsequent discussion.

We chose to do two groups of learning curves analyses, the first using all the data resulting from the models, that is, we transformed the zonal statistic shapefiles into .csv data which served as input for the learning curves analysis available in sklearn. The second group (figure 5) we did the same process, but only with the 40 zonal statistics (representing 10 canopies for each class) used in the in situ validation.

We take the opportunity to describe in more detail the other training and model development steps.

While your expectation that UAV images during wet season might make a difference due the reflectances between the canopy but I don't completely agree on how you came to the conclusion.

Maybe there was some misunderstanding, as this is a result of our study and not a hypothesis to have been expected. In any case, we have changed the text so that there are no misinterpretations.

Further, the text needs to be improved as well. There are many misspellings, and many sentences needs rephrasing. Please correct them as well.

Thanks for the comments, so the text went through a professional translator in order to remedy possible misspellings.

References

De Luca, G., N. Silva, J. M., Cerasoli, S., Araújo, J., Campos, J., Di Fazio, S., & Modica, G. (2019). Object-based land cover classification of cork oak woodlands using UAV imagery and Orfeo ToolBox. Remote Sensing, 11(10), 1238.

Teodoro, A. C., & Araujo, R. (2016). Comparison of performance of object-based image analysis techniques available in open source software (Spring and Orfeo Toolbox/Monteverdi) considering very high spatial resolution data. Journal of Applied Remote Sensing, 10(1), 016011-016011.

CNES OTB Development Team. Software Guide; CNES: Paris, France, 2018; pp. 1–814

OTB Development Team. OTB CookBook Documentation; CNES: Paris, France, 2018; p. 305

Cresson, R.; Grizonnet, M.; Michel, J. Orfeo ToolBox Applications. In QGIS and Generic Tools; John Wiley & Sons, Inc.: Hoboken, NJ, USA, 2018; pp. 151–242

Grizonnet, M., Michel, J., Poughon, V., Inglada, J., Savinaud, M., & Cresson, R. (2017). Orfeo ToolBox: Open source processing of remote sensing images. Open Geospatial Data, Software and Standards, 2(1), 1-8.

Murphy, K. P. (2012). Machine learning: a probabilistic perspective. MIT press.

This manuscript is a resubmission of an earlier submission. The following is a list of the peer review reports and author responses from that submission.

Round 1

Reviewer 1 Report

Congratulations to the authors for the relevant work. They present a relevant method to observe an event that is little studied, especially in the Cerrado. The manuscript is of high quality and can be accepted as is.

The only suggestion is to put the name in English of the Cerrado physiognomies, especially in the figures.

Reviewer 2 Report

Hi,

Please find my comments below:

1. The article seems to compare the different classification methods. However, Kappa Coefficient and Accuracy is not sufficient. Classification Algorithms uses other metrics, such as F1-Scores, AUC, Confusion matrix, etc. Particularly, I would request the Confusion Matrix.

2. I don't understand the need to do SVM, Decision Trees and Random Forests. SVM is one of the oldest algorithms, which is outperformed by DT and RF. However, RF is an ensemble algorithm that uses DT. Comparing DT with RF is like comparing one DT vs many DTs. So, I'm not sure, if this a valid comparison.

3. Since, these algorithms usually take their inputs in a tabular format, I would like to learn more about the data wrangling involved. How, are missing data processed? Did you normalize the values? If so, how? if not, why?

4. It would also be interesting to show, why there were mis-classifications.

5. If the objective is to demonstrate the usability of UAV images, then that can be demonstrated even with one algorithm with an demonstration of what it takes to make the classification more effective. So, unfortunately, comparing multiple algorithms doesn't seem to make sense.

Hope this helps,

Thanks and Regards

Reviewer 3 Report

The work is very similar to "Woody Plant Encroachment: Evaluating Methodologies for SemiaridWoody Species Classification from Drone Images" paper (Olariu et al, 2022), excluding that convolutional networks are not used and that the data pre-processing seems much more complex. They add new non-RGB input channels using multispectral sensor bands (Textural & structural metrics, for example).

In the training they only tell us that "100 points were stipulated for each class". But, there is no specific information about input data structure. Only, in figure 2 a vector output of the segmentation is indicated. What detailed structure have these vectors? Is a k-band vector with their annotated class for every pixel? Or is n,m,k-vector for every annotated class, where n,m is the size of the image an k the number of used bands? So, they should be indicated, as in the article by Olariu et al (2022), which final structure is passed to the 3 classifiers. 

The results that appear in figures 6, 7 and 8 are surprising. Why does DT give a better result than RF with PCA? Since RF is a predictor that averages N DT, it is normal for it to improve the results of 1 single DT.

The exclusion of Convolutional models in problems of artificial vision must be carefully justified. The advantge of use models with low capacity requiriment, as RF or DT, perhaps may doesn't offset the complexity in data pre-traitment.